# Mutations in *GDAP1* Influence Structure and Function of the Trans-Golgi Network

**DOI:** 10.3390/ijms22020914

**Published:** 2021-01-18

**Authors:** Katarzyna Binięda, Weronika Rzepnikowska, Damian Kolakowski, Joanna Kaminska, Andrzej Antoni Szczepankiewicz, Hanna Nieznańska, Andrzej Kochański, Dagmara Kabzińska

**Affiliations:** 1Neuromuscular Unit, Mossakowski Medical Research Centre, Polish Academy of Sciences, 02-106 Warsaw, Poland; kbinieda@imdik.pan.pl (K.B.); wrzepnikowska@imdik.pan.pl (W.R.); akochanski@imdik.pan.pl (A.K.); 2Institute of Biochemistry and Biophysics, Polish Academy of Sciences, 02-106 Warsaw, Poland; damian.kolakowski@ibb.waw.pl (D.K.); kaminska@ibb.waw.pl (J.K.); 3Nencki Institute of Experimental Biology, Polish Academy of Sciences, 02-093 Warsaw, Poland; a.szczepankiewicz@nencki.edu.pl (A.A.S.); h.nieznanska@nencki.edu.pl (H.N.)

**Keywords:** Charcot-Marie-Tooth disease, *GDAP1* gene, Golgi apparatus, inflammation, molecular mechanisms, neuropathy, therapy, yeast-based model organism

## Abstract

Charcot-Marie-Tooth disease (CMT) is a heritable neurodegenerative disease that displays great genetic heterogeneity. The genes and mutations that underlie this heterogeneity have been extensively characterized by molecular genetics. However, the molecular pathogenesis of the vast majority of CMT subtypes remains terra incognita. Any attempts to perform experimental therapy for CMT disease are limited by a lack of understanding of the pathogenesis at a molecular level. In this study, we aim to identify the molecular pathways that are disturbed by mutations in the gene encoding GDAP1 using both yeast and human cell, based models of CMT-GDAP1 disease. We found that some mutations in *GDAP1* led to a reduced expression of the GDAP1 protein and resulted in a selective disruption of the Golgi apparatus. These structural alterations are accompanied by functional disturbances within the Golgi. We screened over 1500 drugs that are available on the market using our yeast-based CMT-GDAP1 model. Drugs were identified that had both positive and negative effects on cell phenotypes. To the best of our knowledge, this study is the first report of the Golgi apparatus playing a role in the pathology of CMT disorders. The drugs we identified, using our yeast-based CMT-GDAP1 model, may be further used in translational research.

## 1. Introduction

Charcot-Marie-Tooth disease (CMT) is a heritable neurodegenerative disease that displays extreme genetic heterogeneity. Over the last three decades, more than 100 genes have been identified as being involved in CMT disease. Despite these advances on the molecular genetics front, the molecular pathogenesis of different CMT subtypes remains unexplored. This lack of understanding at the molecular level has contributed to a lack of effective treatments. CMT disease, resulting from mutations within the gene encoding Ganglioside Induced Differentiation Associated Protein 1 (GDAP1; referred to as CMT-GDAP1), especially from recessive ones, is characterized by a relatively severe clinical outcome. Usually, symptoms first appear during early childhood and the disease is associated with progressive muscle weakness and wasting encompassing distal and proximal parts of the upper and lower limbs. Some patients affected with CMT-GDAP1 are wheelchair-bound [1,2]. To date, more than 100 mutations have been identified in *GDAP1* from patients exhibiting a CMT phenotype. The vast majority of them are inherited as autosomal recessive traits, however dominant mutations are also described [3].

*GDAP1* was first identified in 1999 as one of a series of ten transcripts, the expression of which are highly elevated upon ganglioside-induced differentiation of murine neuronal cell lines [4]. In 2002, two independent groups identified mutations in *GDAP1* that led to the rare polyneuropathy, CMT disease [5,6]. Over the last 20 years, knowledge relating to the structure and function of GDAP1 has increased significantly. However, many aspects of its role in cell function and, in particular, of the pathogenesis of CMT-associated *GDAP1* mutations are still not understood [3,7]. This lack of understanding hampers efforts to develop an efficient therapy. Currently, there is no efficient treatment for CMT-GDAP1 and, furthermore, no single substance has reached clinical or even pre-clinical testing.

GDAP1 is an outer-mitochondrial membrane protein [8,9] with primary sequence homology to members of the glutathione S-transferase (GST) superfamily [6,10]. However, its activity as a GST is highly questionable [9,11,12]. The GDAP1 protein is involved in many aspects of mitochondrial physiology [3,7]. In addition, a number of studies suggest that GDAP1 with the junctophilin-1 is involved in the regulation of calcium homeostasis in cells [13,14,15]. It is therefore interesting that heterologous expression of *GDAP1* in yeast suppresses the calcium hypersensitivity of a strain that lacks *CSG2* gene (*csg2*Δ), encoding the protein required for the formation of complex sphingolipids [16]. It was also reported that GDAP1 is targeted to peroxisomes, where it regulates their morphology [17].

Impairment of mitochondrial homeostasis is seen as the major cause of the CMT-GDAP1 disease. Phenotypes associated with a *GDAP1* mutation include: disruption of mitochondrial fission-fusion events; changes in mitochondrial distribution; impairment of the mitochondrial transmembrane potential; an increase in reactive oxygen species levels; a reduction in glutathione content; and an alteration of mitochondrial bioenergetics [8,18,19,20,21,22,23]. Additionally, expression of the mutated *GDAP1-Gly327Asp* and *GDAP1-Leu239Phe* alleles in yeast cells increases the rate of DNA escape from mitochondria to the nucleus, indicating an abnormality in mitochondrial functioning [16]. Recently, Fernandez-Lizarbe and coworkers have used the *GDAP1* knockout (*GDAP1* -/-) mouse model to show that inflammation in the spinal cord and sciatic nerve contributes to the pathophysiology of *GDAP1*-related CMT [24].

The Golgi apparatus (henceforth referred to simply as the Golgi) is a membrane-bound organelle that plays a central role in the trafficking, processing, and sorting of proteins and lipids. It is often located adjacent to the nucleus in vertebrate cells and consists of several flattened cisternae piled on top of each other and aligned in parallel into compact stacks. Tubular membranes interconnect these stacks to form a ribbon-like network. However, in certain unicellular eukaryotes, including the yeast *Saccharomyces cerevisiae*, the Golgi structures appear as single, isolated cisternae randomly distributed throughout the cytoplasm [25,26]. The Golgi is highly polarized; proteins and lipids from the endoplasmic reticulum (ER) are transported to the *cis* compartment of the Golgi, then exported through *medial-* to the *trans*-Golgi. In the *trans*-Golgi network (TGN), the cargo is separated, sorted and targeted to other cellular structures including the endosomes, lysosomes and plasma membrane or exported outside of the cell [27]. When passing through the Golgi compartments, cargo molecules are modified and processed. Individual sub-compartments of the Golgi stacks contain unique sets of resident proteins, responsible for maintaining the architecture and function of the Golgi and for modification of cargoes [26,27].

To the best of our knowledge, the present work shows for the first time that, in addition to its roles in mitochondrial and peroxisomal function, GDAP1 also influences the structure and probably the functioning of the Golgi. Decreased levels of GDAP1 protein in SH-SY5Y and HeLa cell lines results in changes in the Golgi morphology as observed by transmission electron microscopy (TEM). What is more, the reduction of the native levels of GDAP1 in HeLa cells alters the maturation and localization of the *trans*-Golgi protein TGN46. In yeast cells, expression of the *GDAP1-Leu239Phe* gene specifically affects the distribution of *trans*-Golgi protein Sec7, suggesting changes in the organization of the Golgi network and demonstrating that yeast is a convenient model in which to study CMT-GDAP1.

To further dissect the role of GDAP1 in the cell and especially how it influences Golgi function, we used a yeast GDAP1-model. Firstly, we investigated the mechanism by which the calcium hypersensitivity associated with *csg2*Δ strain is suppressed by human *GDAP1*. We have found that *GDAP1*-mediated suppression of the calcium sensitivity of *csg2*Δ does not require ongoing sphingolipid synthesis. This effect is not dependent on normal functioning of the Golgi, as expression of *GDAP1-Leu239Phe*, which results in alterations to the Golgi network in yeast, is as beneficial as the expression of wild type *GDAP1*. Secondly, we decided to screen a drugs library for substances that, similarly to *GDAP1*, suppress the grown defect of *csg2*Δ on calcium-containing medium. The drugs that exhibited a suppressive effect suggest that the effects of GDAP1 on cell function are mediated via different pathways. In summary, we have shown that *GDAP1* mutations affect the structure and probably the functioning of different molecular pathways that affect the TGN.

## 2. Results

### 2.1. GDAP1 Protein Levels Influence Mitochondria and Golgi Morphology in SH-SY5Y Cells

The molecular function of GDAP1 in cells has not been determined. Here we investigated the effect of altering *GDAP1* expression levels (either reduced or overexpressed) in the SH-SY5Y cell line derived from human neuroblastoma. Neuron-derived cells are a suitable model to study *GDAP1*-associated effects due to the involvement of *GDAP1* in the pathogenesis of peripheral neuropathies. To reduce expression of *GDAP1*, RNA silencing was used. Western blot analysis revealed that, 24 h after siRNA transfection, the level of GDAP1 was markedly reduced in comparison to the control. This reduction of expression could still be observed, albeit less so, 48 h post-transfection (Figure 1a).

It can also be seen that overexpression of *GDAP1* was very effective (Figure 1a). The influence of GDAP1 concentration on the structure of the mitochondria and Golgi was investigated using transmission electron microscopy (TEM).

It was previously shown that GDAP1 is involved in the maintenance of mitochondrial morphology. Thus, we began by analyzing the general appearance of these organelles in cells with altered levels of GDAP1. Figure 1b shows that overexpression of *GDAP1* affects mitochondrial size and shape: mitochondria were smaller and rounder compared to those in the control cells. This finding is in agreement with previously published data [8,28]. The effect of *GDAP1* gene silencing on mitochondria was not evident. We cannot discriminate between cells without and with significant *GDAP1* silencing (Figure 1a). We were unable to determine GDAP1 levels for individual cells and thus could not associate mitochondrial morphology with a specific GDAP1 concentration. Surprisingly during the examination of the TEM images, we found that even a partial reduction of GDAP1 levels resulted in alterations to Golgi morphology; the cisternae were more irregular in shape and the ends of the stacks are denser (Figure 1d). Thus, TEM analysis of a neuroblastoma cell line suggested that GDAP1 protein may also influence the structure and function of the Golgi.

### 2.2. Exogenous Expression of Particular GDAP1 Variants Affects the Endogenous GDAP1 Protein Levels in HeLa Cells

As the previous experiment pointed to the involvement of GDAP1 in maintenance of the Golgi structure, we hypothesized that the pathogenic effects of *GDAP1* mutations may, at least partially, result from a failure of this organelle. To test this hypothesis, we examined the influence of selected *GDAP1* variants on Golgi morphology. The SH-SY5Y cell line is characterized by a high expression of endogenous *GDAP1*, which makes it impossible to observe the effects of recessive *GDAP1* gene variants. Thus, for further experiments, we used the HeLa cell line that was previously used to characterize *GDAP1* mutants [12,21,29]. Six *GDAP1* mutations were analyzed (Figure 2a) based on the following criteria: (i) whether the *GDAP1* mutation is inherited as an autosomal recessive (4 alleles) or autosomal dominant trait (3 alleles), (ii) frequency of the mutations (*GDAP1-Leu239Phe*) in the population (recurrent mutation) and (iii) availability of clinical data for phenotypes of CMT disease associated with a specific mutation. The *GDAP1-Pro153Argfs*19* mutation has never been identified in a patient with CMT; it was generated by mutagenesis and was included here to represent the frame-shift, loss of function mutations of *GDAP1*.

To study the effect of expressing selected *GDAP1* alleles in HeLa cells, we initially analyzed the levels of mutant GDAP1 proteins. Surprisingly, it was observed that following transfection with vector only, or with vector bearing wild-type *GDAP1* or with specific *GDAP1* alleles (namely *GDAP1-Pro153Argfs*19*, *GDAP1-Gln218Glu* and *GDAP1-Leu239Phe*) the total level of GDAP1, including the endogenously expressed protein, dropped markedly (Figure 2b). However, following expression of the *GDAP1-Gly327Asp*, *GDAP1-Glu222Lys* and *GDAP1-His123Arg* alleles, the level of GDAP1 protein increased to a level comparable to that observed for the SH-SY5Y cell line (Figure 2b). Based on this observation, we concluded that *GDAP1* variants may be assigned into two groups. The first group displays the same effect observed for expression of wild type *GDAP1*—silencing of the endogenous *GDAP1*. The second group includes certain variants, expression of which resulted in the overproduction of *GDAP1*.

### 2.3. A Reduction of GDAP1 Expression Alters Golgi Morphology in HeLa Cells

Expression of some *GDAP1* variants in HeLa cells clearly leads to a reduction of total GDAP1. We checked if, similarly to neuroblastoma cells, a decreased amount of GDAP1 in HeLa cells also resulted in alterations to the Golgi morphology. TEM analysis revealed that intact Golgi were visible in about 20% of control cells. Approximately 50% of the control cells exhibited an altered Golgi morphology; one that takes the form of cisternae plus vacuoles (altered type I). The fraction of cells—in which the Golgi in the form of cisternae is invisible—was also recorded (altered type II). In cells expressing the *GDAP1-Leu239Phe* allele (with a reduced GDAP1 level) the number of cells with distinguishable cisternae (both unaltered and altered type I) decreased while the percentage of cells in which the Golgi took the form of vacuoles only (altered type II) increased (Figure 3). The cells expressing *GDAP1-Gly327Asp* (resulting in overexpression of *GDAP1*) also exhibited an altered Golgi morphology, but the changes were minor compared to those observed in cells expressing *GDAP1-Leu239Phe*. Expression of *GDAP1-Gly327Asp* resulted in a slight increase in the number of cells in which the Golgi took the form of vesicles (Figure 3).

### 2.4. A Reduction of GDAP1 Expression Results in Changes to Post-Translational Modifications of TGN46

We asked if alterations of the Golgi structure in HeLa cells is also manifested at a molecular level as changes in the levels or modifications of Golgi proteins. Initially we analyzed the levels of Golgi proteins that are characteristic of the *cis*- and *trans*-Golgi regions. There were no statistically significant changes in the level of the *cis* Golgi marker, GM130, but changes in the pattern of bands of the *trans*-Golgi marker, TGN46, were observed (Figure 4).

TGN46 is a type I transmembrane glycoprotein, which migrates as several bands during electrophoresis. The >110 kDa form is the mature sialylated protein. Western blot analysis revealed that the expression of some *GDAP1* mutant alleles resulted in a drop in the endogenous GDAP1 level, and this correlates with a change to the pattern of posttranslational modification of TGN46. In cells expressing *GDAP1, GDAP1-Pro153Argfs*19, GDAP1-Gln218Glu* and *GDAP1-Leu239Phe*, TGN46 did not fully mature; the slower migrating forms, shown previously to be sialylated forms, were absent. A 70 kDa form was more pronounced in the cells expressing the mutant alleles (Figure 4). Thus, a reduction in GDAP1 levels alters some processes carried out in the Golgi, reflected by changes in TGN46 maturation.

### 2.5. Localization of TGN46 Is Altered in HeLa Cells Expressing the GDAP1-Leu239Phe Variant

Post-translational modification of proteins regulates their activity, stability and localization. As discussed above, HeLa cells with a reduced level of GDAP1 exhibited alterations in expression and maturation of TGN46. We therefore decided to test if these alterations also affected the localization of TGN46 in cells with expression of two chosen mutant alleles as the representatives of two different *GDAP1* expression level. These mutations are the most common one, *GDAP1-Leu239Phe,* and *GDAP1-Gly327Asp*, which results in disturbing GDAP1 targeting to the mitochondrial membrane [2,30]. In order to visualize the morphology of the Golgi we immunostained the GM130, GORASP2 and B4GALT3 proteins. All proteins were observed using confocal microscopy. In the control HeLa cells, GM130 ( *cis*-Golgi marker), GORASP2 (*medial*-Golgi), B4GALT3 (TGN) and TGN46 (TGN), were present in structures near one pole of the nucleus, as expected for Golgi proteins. A similar localization of these proteins was observed for cells expressing the *GDAP1-Gly327Asp* allele. In contrast, in cells expressing *GDAP1-Leu239Phe*, the distribution of TGN46 was altered; it was localized in tubular structures (Figure 5). This mislocalization of TGN46 could arise as a consequence of altered post-translational modification or, conversely, the mislocalization could be responsible for alterations in post-translational modification of TGN46.

### 2.6. Yeast Cells Transformed with the GDAP1-Leu239Phe Gene Allele Exhibit Changes in the Localization of Sec7

Recently, we have demonstrated that it is possible to determine the effects of *GDAP1* expression on yeast cell physiology. In addition we showed that expression of different *GDAP1* alleles may have different effects on the functioning of yeast cells [16]. We hypothesized that the effects of a *GDAP1* variant on the Golgi were similar for both the HeLa and yeast cell models. Thus, we decided to check if expression of the same *GDAP1* alleles as in HeLa cells: *GDAP1*, *GDAP1-Gly327Asp* and *GDAP1-Leu239Phe* could alter the localization of Golgi proteins in yeast cells. In order to compare models, we monitored two different Golgi marker proteins, Sed5 and Sec7, tagged with red fluorescence protein (mRFP). Sed5 is a *cis*-Golgi marker while Sec7 serves as a TGN marker [31]. In yeast cells, the Golgi structure differs from the classical cisternae stacks observed in mammalian cells. Yeast Golgi takes the form of dispersed cisternae throughout the cytoplasm, observed as single puncta. Confocal microscopy revealed that the number of Sed5-mRFP spots was similar in all examined transformants (Appendix A). In contrast, cells expressing *GDAP1-Leu239Phe* had a higher number of Sec7-mRFP containing puncta (Figure 6). This suggests that the yeast *trans*-Golgi could also be affected by expression of *GDAP1-Leu239Phe*. As the phenotypes observed in HeLa cells are similar to those observed for yeast cells, it raises the possibility that yeast cells could be used to further investigate the influence of *GDAP1* variants on Golgi structure and function.

### 2.7. GDAP1-Mediated Suppression of the Calcium Sensitivity of csg2*Δ* Does Not Require Ongoing Sphingolipid Synthesis

The effect of *GDAP1* gene expression on yeast cell physiology is apparent when it is expressed in a *csg2*Δ mutant. This expression results in suppression of the calcium sensitivity of the *csg2*Δ mutant and different *GDAP1* variants either retain or lose this suppressive ability [16]. Csg2 is a Golgi localized protein, which is required for mannosylation of inositolphosphorylceramide i.e., the formation of complex sphingolipids (Figure 7a).

It was shown that Csg2 is also required for the efficient retention of some *medial*-Golgi enzymes [32]. To uncover the molecular mechanisms underlying *GDAP1*-mediated suppression of calcium sensitivity in a *csg2*Δ mutant, and to test if the observed effect is Golgi-dependent, we asked if ongoing sphingolipid synthesis is necessary for the efficiency of *GDAP1*-mediated suppression. Furthermore, we asked if the *GDAP1-Leu239Phe* variant, which affects *trans*-Golgi protein localization in yeast, improves the growth of the *csg2*Δ mutant in the presence of calcium ions. We examined if the addition of myriocin, which blocks the first enzyme of the sphingolipid synthesis pathway, abolishes *GDAP1*-mediated suppression. To answer these questions the growth of the *csg2*Δ mutant on plates containing calcium ions was compared to growth on plates containing calcium ions and myriocin. The growth of *csg2*Δ was reduced by the addition of calcium ions and further by the addition of myriocin. When *csg2*Δ was transformed with *GDAP1* or the *GDAP1-Leu239Phe* allele, it grew better than when transformed with an empty plasmid or with the *GDAP1-Gly327Asp* allele (Figure 7b). This suggests that the presence of GDAP1 protein reduces the toxicity of calcium ions in a *csg2*Δ mutant in a manner that does not involve ongoing sphingolipid biosynthesis. Furthermore, this suggests that the *GDAP1*-based suppression is not related to the effects of the GDAP1 protein on the Golgi; the expression of *GDAP1-Leu239Phe*, which changes the modification and localization of TGN46 in HeLa cells and number of Sec7-mRFP containing structures in yeast cells, has a positive effect on the growth of the *csg2*Δ mutant, which already has an altered Golgi apparatus.

### 2.8. Screening a Drug Library Using the Calcium Hypersensitivity of the csg2*Δ*Strain Revealed Ibuprofen Piconol as an Active Compound

We have already ruled out the influence of GDAP1 protein on the sphingolipid biosynthesis pathway and related GDAP1’s effects to changes in the structure of the Golgi. The molecular mechanism by which GDAP1 acts in *csg2*Δ cells is still unknown. For a further dissection of the *GDAP1*-dependent suppression of calcium hypersensitivity in a *csg2*Δ mutant, we used a chemical suppressor screen, assuming that we would find a compound that exerts an effect similar to that caused by the expression of *GDAP1*. Identification of such a compound, coupled with its mechanism of action, will be used to determine the role of GDAP1. We adapted the Ca^2+^ ion sensitivity phenotype of a *csg2*Δ mutant for use in a drug screen (Figure 8), in order to identify molecules that suppress the calcium hypersensitivity associated with *csg2*Δ. We screened the Prestwick drug repurposing library (Prestwick Chemical Library) previously used in such screens [33,34,35], which contains drugs approved for use in humans.

Library compounds were applied to filter discs on top of solid media. Active compounds were identified after 3–5 days by observation of growth zones around the filter. Two drugs were identified as suppressors of the *csg2*Δ growth defect on Ca^2+^ ion containing media. The first was a previously identified chemical suppressor: cyclosporine A. The second was ibuprofen piconol (Figure 8), a nonsteroidal anti-inflammatory agent that inhibits synthesis of prostaglandins in mammalian cells, but is also regarded as a phospholipase inhibitor. In summary, expression of *GDAP1* suppresses *csg2*Δ, not through biosynthesis of sphingolipids, but via another conserved pathway, most probably connected with calcium homeostasis, inhibition of phospholipase A or both.

## 3. Discussion

Although almost 20 years have passed since *GDAP1*-associated CMT disease was first described, its molecular pathogenesis is still not understood. The molecular pathogenesis of CMT-GDAP1 disease has been linked to a mitochondrial pathology and to disturbances in calcium homeostasis. The uncertainty as to the underlying causes of the pathogenesis mean that even experimental therapies are not possible. Clinical trials for the treatment of patients suffering from other types of CMT have been conducted and published, which suggests a road map for the future [36,37]. The main issue here is the diversity of phenotypes caused by different mutations in *GDAP1*. In the present study, we have shown two additional molecular phenotypes associated with mutations in *GDAP1*. Here we analyzed six mutations in *GDAP1*, three resulted in lower levels of GDAP1, and the other three did not. In addition, the mutations that influence GDAP1 levels, such as *GDAP1-Leu239Phe*, also affected the morphology of the Golgi. These alterations in morphology are visualized as a conversion of Golgi structures (from cisternae to vesicles) as detected by TEM in neuroblastoma as well as in HeLa cells. These changes are accompanied by changes in the modification and localization of TGN46, a protein of the TGN. The impact of GDAP1 on the Golgi was also apparent in yeast cells expressing *GDAP1-Leu239Phe*. We then used yeast model to screen for drugs that could restore normal functioning in the absence of GDAP1. To the best of our knowledge, this study is the first report of a link between disturbances in the Golgi and the molecular pathogenesis of Charcot-Marie-Tooth disorders.

Initial studies suggested that *GDAP1* expression was limited to the neural cell lines. Later studies suggested that this is not the case: human fibroblasts contain about 2.6% *GDAP1* mRNA levels found in motor neurons and at least some *GDAP1* mutations result in a dramatic reduction of *GDAP1* mRNA expression in CMT patients fibroblasts [20]. Interestingly, the same effect, the very low levels of GDAP1, is observed in the fibroblasts of patients carrying the homozygous *GDAP1-Leu239Phe* and *GDAP1-Arg273Gly* mutations [20] and in HeLa cells transfected with a plasmid expressing the *GDAP1-Leu239Phe* variant. This similarity between the molecular effects of the *GDAP1-Leu239Phe* mutation in two divergent cell lines (fibroblasts biopsied taken from patient and HeLa cells) supports our use of HeLa cells in these studies. Thus, mutations that reduce GDAP1 levels have a similar effect to loss of function mutations or even knockout mutations. It is interesting to establish how the level of GDAP1 correlates with the severity of clinical manifestation. At least in other CMT type diseases caused by mutations in the *IGHMBP2* gene, there is a correlation between the IGHMBP2 level and severity of the disease [38].

The TGN is a particularly dynamic structure: microscopic observations show the continuous formation of tubules and vesicles in this part of the Golgi. This process requires tubulin dimers. Tubules appear to be drawn from the TGN along the microtubules. Once formed, the vesicular tubular carriers move towards the periphery of the cell, transported along the microtubules [39]. It is proposed that the defects in microtubule-dependent trafficking lead to neurodevelopmental and neurodegenerative diseases including Alzheimer’s disease, Parkinson’s disease, amyotrophic lateral sclerosis (ALS; also known as Motor Neurone Disease or Lou Gehrig Disease) or traumatic brain injury. Recent studies highlight the use of microtubule stabilizing agents as potential drugs for improving axonal transport and therefore nerve function in these diseases [40]. Noteworthy, the pathogenesis of CMT2Z caused by mutations in *MORC2* gene could be also linked to disturbances in trafficking along microtubules [41]. Beta-tubulin is known to interact with GDAP1 [14,15], therefore changes in the interaction between beta-tubulin and GDAP1, caused by mutations in *GDAP1*, could impact microtubules and subsequently TGN structure and function. However, the changes caused by mutations in *GDAP1* are subtle and impact only the TGN. In contrast, for Alzheimer’s disease, Parkinson’s disease and ALS, fragmentation of the whole Golgi has been reported [42]. The number of reports showing fragmentation of the Golgi limited to the TGN is scarce. In HEK293 cells expressing Rab71 and in a LRRK2 mutant (a model for Parkinson’s disease), Golgi fragmentation was limited to the TGN with the *cis-* and *medium*-Golgi segments remaining intact [43]. A similarly specific TGN fragmentation was observed for Vero76 cells infected with *Rickettsia rickettsii* [44]. In Parkinson’s disease, the Golgi dispersion was mediated by Rab7L1 GTPase while in the *Rickettsia* infected cells, dispersion was mediated by RARP2 (Rickettsial Ankyrin Repeat Protein 2) suggesting heterogeneous molecular mechanisms for TGN disruption [43,44,45]. Moreover, recent studies have demonstrated that the Golgi acts as a platform for the formation of signaling complexes triggering the inflammatory response [46] and that the disassembly of TGN is a crucial event in the activation of the NRPL3 inflammasome. Diverse NRPL3 activators trigger this response, but the *cis-* and the *medial*-Golgi remain intact throughout [47]. In the present study, we have shown for the first time that the TGN may also be disrupted in CMT disease or at least in some CMT subtypes caused by mutations within *GDAP1*. We have revealed this in a series of three independent experiments encompassing neuroblastoma, HeLa and yeast cells. Similarly to the previously reported studies in which selected TGN disruption was observed, the Golgi disturbance we detected in HeLa cells transfected with *GDAP1-Leu239Phe* was limited to the TGN only. Additionally impaired function of the TGN, at the very least, results in changes to the glycosylation pattern and localization of TGN46. Meanwhile, the localization patterns of *cis-*, *medial-* and other TGN proteins (GM130, GORASP2 and B4GALT3, respectively) remain intact, unchanged from the pattern observed for control cells. Our finding that the Golgi is affected in CMT sheds new light on the pathology of CMT-GDAP1, which, so far, has only been associated with mitochondrial dysfunction encompassing mitochondrial fission/fusion processes; mitochondrial transport and impaired oxidative phosphorylation [3]. How some *GDAP1* gene mutations may selectively affect TGN46, but not B4GALT3 in the TGN of HeLa cells remains unclear. However, there is evidence that the TGN is organized into distinct sub-compartments, marked by specific TGN proteins. Examples of the specific localization of proteins include the distinct localizations of ATP7A, a copper transporting ATPase, and TGN46 [48]. Another example is the localization of golgins: proteins that function as molecular tethers during the docking of transport vesicles to a target membrane during the maintenance of stacked cisternae structures in the Golgi. Two such golgins, GCC88 and GCC185, localize to a common TGN sub-compartment. Alpha-2, 6-sialyltransferase also localizes to this sub-compartment but TGN46 is absent. This sub-compartment is distinct from that which contains p230/golgin-245 and golgin-97 [49]. The mechanism/pathway by which GDAP1 influences the modification and localization of TGN46 remains to be studied. It is also possible that mutations in the *DNM2* gene encoding a dynamin and causing CMTIB may affect the structure of the TGN as well as they affect the receptors endocytosis [50]. The same may be true for mutations in the *SH3TC2* gene causing CMT4C. These mutations also affect endocytosis, which may reflect a general disturbance of intracellular transport from the TGN and in later stages [51]. To find the molecular mechanism leading to changed TGN46 localization, the molecular partners of GDAP1 need to be identified. From our studies, we predict that the molecular pathway by which GDAP1 influences TGN structure and function is conserved. Similarly to what was observed in HeLa cells, the expression of *GDAP1-Leu239Phe* in yeast cells changed the distribution of Sec7-mRFP (a TGN marker) while localization of Sed5 (a *cis*-Golgi marker) remained unchanged. This conservation allows us to use the yeast model to identify the molecular partners of GDAP1 and to uncover the pathways that influence TGN functioning.

The *GDAP1* mRNA is one of 10 mRNAs that are highly expressed in the Neuro2a cell line (a mouse neuroblastoma cell line) following neurite-like differentiation. Differentiation is achieved by transfection of these cells with a cDNA encoding the gene for the GD3 synthase [4]. The GD3 synthase is a Golgi membrane protein that catalyzes the formation of GM3 ganglioside, a ceramide-based glycolipid. In yeast, the sphingolipid composition is quite simple: yeast contain three inositol-containing complex sphingolipids only with one mannose. Csg2 influences the mannosylation of inositolphosphorylceramide [52]. Here we present evidence that the effect of *GDAP1* expression in a *csg2*Δ mutant is most likely not related to its effect on the complex sphingolipid biosynthetic pathway and, by extension, on Golgi structure. Two lines of evidence from our yeast study suggest such a conclusion. Firstly, the addition of myriocin, which inhibits the first enzyme of the sphingolipid biosynthesis pathway, does not abolish the suppression of a *csg2*Δ mutant through the expression of *GDAP1*. Secondly, a comparison of the effect of *GDAP1* expression with the known action of chemical suppressors, isolated in a screen, suggests an indirect effect. The identification of cyclosporine A in the drug screen suggests the involvement of calcium homeostasis. It has been shown that *GDAP1*-knockouts in either human neuroblastoma SH-SY5Y cells or in mice motor neurons exhibit a defect in store-operated calcium entry (SOCE), a calcium cell-entry pathway [14,15]. This phenotype is linked to an observed mislocalization of mitochondria [14]. The selection of ibuprofen as a chemical suppressor mimicking the effect of *GDAP1* expression in the *csg2*Δ mutant suggests that there may be an additional mechanism. Ibuprofen is the most interesting candidate drug due to the recent findings of Fernandez-Lizarbe et al. regarding neuroinflammation in *GDAP1*-knockout mice [24]. The authors of this elegant study have demonstrated the presence of elevated levels of some proinflammatory mediators (TNF-α and pERK, and the C1qa and C1qb proteins of the complement system) supporting the role of inflammation in the pathogenesis of CMT-GDAP1 disease. Ibuprofen is an anti-inflammatory agent that is able to restore growth of the *csg2*Δ mutant yet there are no inflammatory processes in yeast cells. This suggests that the initial stages (i.e., biochemical pathways) of the inflammatory process are present in yeast cells. We speculate that this process is related to the maintenance of TGN structure. Regardless of the mechanism of action of ibuprofen, the presence of an inflammatory response in *GDAP1*-/- mice classify this compound as one of the first potential drugs with which to treat CMT-GDAP1 disease. Given the relatively low number of adverse effects associated with ibuprofen, its use in experimental CMT-GDAP1 therapy is quite realistic.

In conclusion, our work shows that the mitochondrial protein GDAP1 involved in the pathology of CMT is associated with changes in a distinct cell compartment, the TGN. These changes could be responsible for inducing inflammation. Drugs with different mechanisms of action were shown to have activity in the yeast CMT-GDAP1 model. This indicates that multiple molecular pathways can compensate for the lack of GDAP1. Finally, ibuprofen is an encouraging candidate for the treatment of CMT-GDAP1 disease.

## 4. Materials and Methods

### 4.1. Yeast Strains, Media and Growth Conditions

The yeast *Saccharomyces cerevisiae* strains used in this study were BY4741 MATa *his3*Δ1 *leu2*Δ*0* met15Δ*0 ura3*Δ*0* and BY4741 *csg2*Δ. Yeast cultures were grown at 28 °C or at room temperature in YPD medium (1% yeast extract, 2% peptone, 2% glucose) or in complete synthetic medium (SC) (0.67% yeast nitrogen base with ammonium sulfate without amino acids, 2% glucose with complete supplement mixture). For growth tests, the optical cell density (OD_600_) was determined and cultures were diluted with SC-leu medium to OD_600_ ≈ 1. Aliquots of 10-fold serial dilutions of cells were spotted on solid YPD media plates supplemented as indicated. Plates were incubated at room temperature for the number of days indicated.

Neuroblastoma cell line SH-SY5Y (gift from M.Szeliga MMRC PAS) was maintained in DMEM/F12 (Sigma-Aldrich, Saint Louis, MI, USA) containing 10% FBS (Thermo Fisher Scientific, Waltham, MA, USA), 100 IU/mL penicillin, and 100 μg/mL streptomycin (Thermo Fisher Scientific), + 1% Non-Essential Amino Acids (NEAA) (Sigma-Aldrich) and uridine 1mg/mL (Sigma-Aldrich). The HeLa cell line from American Type Culture Collection (ATCC, Manassas, VA, USA) was grown in Dulbecco’s modified Eagle’s medium (DMEM high glucose, GlutaMAX, Thermo Fisher Scientific) containing 10% FBS (Thermo Fisher Scientific), 100 IU/mL penicillin, and 100 μg/mL streptomycin (Thermo Fisher Scientific) and uridine 1mg/mL (Sigma-Aldrich) in a humidified tissue culture incubator at 37 °C and 5% CO_2_ atmosphere.

For transmission electron microscopy analysis (TEM), SH-SY5Y and HeLa cells were cultured at the seeding density of 1 × 10^5^ cells on Nunc Thermanox Coverslips (ø13 mm, Thermo Fisher Scientific, nr cat.: 174950), placed into 24-well plates and incubated for 24 h until they reached approximately 70–80% confluency.

### 4.2. Cell Transfection and GDAP1 Silencing

HeLa and SH-SY5Y cells were transfected using the pIRES2-AcGFP1 Vector with Viromer Red (Lipocalyx, Halle, Germany) according to the manufacturer’s recommendations with the standard transfection scale. In all cases, the medium was changed 48 h after transfection. The resulting transfectant HeLa and SH-SY5Y cells were cultured in an appropriate medium with 500 μg/mL G418 (Lab Empire, Rzeszów, Poland) for 2 weeks.

For the generation of a *GDAP1*-silenced SH-SY5Y cell line, cells were seeded on 24-well plate to a confluency of about 70–80% and transfected using 1.6 μL of siRNA (10 µM solution) (Qiagen, Valencia, CA, USA) per well and Lipofectamine 3000 (Thermo Fisher Scientific) as recommended by the manufacturer. Clonal cell lines were tested for *GDAP1* silencing by Western blot.

### 4.3. Plasmids and DNA Manipulations

The plasmids used in this study are listed in Table 1.

*GDAP1* cDNA was subcloned from pCMV6-XL5 GDAP1 (NM_018972) Human Untagged Clone (OriGene Technologies Inc., Rockville, MD, USA) to pIRES2-AcGFP1 Vector (TAKARA Bio, Shiga, Japan) using SacI and SalI enzymes (Thermo Fisher Scientific).

Mutagenesis for mutations in the *GDAP1* gene (c.980G > A p.Gly327Asp, c.458C > T p.Pro153Leu, c.652C > G p.Gln218Glu, c.664G > A p.Glu222Lys, c.715C > T p.Leu239Phe and c.368A > G p.His123Arg) were performed using Mut Express II Fast Mutagenesis Kit V2 (Vazyme Biotech Co., Ltd., Nanjing, Jiangsu, China) according to the manufacturer’s instructions and verified by sequencing.

### 4.4. Protein Extracts and Western Blot Analysis

Total protein cell extracts were prepared using a cell lysis buffer (Cell Signaling Technology, Inc., Danvers, MA, USA) with 1 mM PMSF and Protease Inhibitor Cocktail 1/200 *v*/*v* (Sigma-Aldrich). Protein were separated by SDS-PAGE, transferred onto nitrocellulose membrane Amersham Protran (GE Healthcare Bio-Sciences AB, Uppsala, Sweden) and analyzed by standard Western blotting using rabbit polyclonal anti-GDAP1 (Abcam, Cambridge, MA, USA), purified mouse anti-GM130 (BD Transduction Laboratories, San Jose, CA, USA), rabbit polyclonal anti-B4GALT3 (Proteintech, Chicago, IL, USA), rabbit polyclonal anti-GORASP2 (Proteintech), rabbit anti-TGN46 (Sigma-Aldrich), mouse monoclonal anti-beta Actin (Proteintech) antibodies and secondary anti-rabbit IgG and anti-mouse IgG horseradish peroxidase (HRP)-conjugated antibodies (Sigma-Aldrich) followed by enhanced chemiluminescence (Western Bright Sirius Advansta, San Jose, CA, USA).

### 4.5. Drug Screening Assay

The assay was based on previously described tests [33,34]. Cells of the yeast *S. cerevisiae* strain *csg2*Δ were grown to early exponential phase, OD_600_ was adjusted to 0.5, and 200 μL was spread homogenously onto solid YPD media supplemented with 0.5 M CaCl_2_. Sterile filters were placed on the media surface and 5 μL of 10 mM drug solutions in dimethyl sulfoxide (DMSO) were applied to each filter disc. In this screen, all 1520 compounds from the Prestwick Chemical Library (Prestwick Chemical Libraries, Illkirch-Graffenstaden, France), 99% of which have been approved by Food and Drug Administration, USA, were tested. DMSO was used as a vehicle control. As a positive control the *csg2*Δ strain with plasmid encoding human *GDAP1* gene (*csg2*Δ [*GDAP1*]) was used. The *csg2*Δ [*GDAP1*] strain was spotted directly on the surface of the media in the corner of each plate. The plates were incubated at 28 °C for 3 to 5 days. Data was recorded using a Snap Scan1212 (Agfa, Mortsel, Belgium).

### 4.6. Confocal Microscopy

HeLa and SH-SY5Y cells were grown on 12 mm poly-L-lysine-coated coverslips. Cells were fixed by incubation for 15 min in 4% formaldehyde in PBS followed by 10 min of permeabilization in 0.5% Triton-X in PBS and then blocked for 1 h in 0.1% Triton-X and 2% BSA in PBS. The slides were incubated overnight with primary antibody: purified mouse anti-GM130 diluted 1:40 (BD Transduction Laboratories), rabbit polyclonal anti-B4GALT3 diluted 1:65 (Proteintech), rabbit polyclonal anti-GORASP2 diluted 1:300 (Proteintech), rabbit anti-TGN46 diluted 1:200 (Sigma-Aldrich). After washing, slides were incubated with secondary antibody Alexa Fluor 546-conjugated goat anti-rabbit or anti-mouse IgG (Invitrogen, Camarillo, CA, USA) diluted 1:500 for 1 h. Then slides were stained using DAPI (Thermo Fisher Scientific) for 15 min and mounted in mounting medium (DAKO, Agilent, Santa Clara, CA, USA).

To observe the morphology of Golgi in yeast cells, BY4741 strains carrying a *SEC7-mRFP*- or *SED5-mRFP*- containing plasmid were transformed with empty plasmid or plasmids bearing the indicated *GDAP1* variants. Strains were grown overnight in SC-leu-ura media at 28℃. Cultures were diluted (1:10) in fresh SC-leu-ura medium for 4 h. Cells were collected by centrifugation, fixed by incubation for 25 min in 4% formaldehyde and washed in KiPO4/sorbitol buffer (100 mM potassium phosphate pH 7.5 and 1.2 M sorbitol). Fixed cultures were spotted on poly-L-lysine- or concanavalin A- coated slides and the nucleus was stained by DAPI (Thermo Fisher Scientific) for 2 min. Slides were then washed with water and mounted in a mounting medium (DAKO).

Cells were viewed using LSM 780 Axio Observer Z.1 confocal microscope (Zeiss, Oberkochen, Germany). Images were collected using Zen 2012 black edition software (Zeiss). The confocal microscopy observations were performed in the Laboratory of Advanced Microscopy Techniques, Mossakowski Medical Research Centre, PAS. At least 100 yeast cells with Sec7-mRFP and Sed5-mRFP were counted for every experimental variant.

### 4.7. Transmission Electron Microscopy

Cells were fixed by 2.5% glutaraldehyde, 2% paraformaldehyde (Electron Microscopy Sciences, Hatfield, PA, USA) solution for 1 h at 4° C. After fixation, cells were rinsed three times for 10 min in 0.1 M cacodylate buffer (BDH Chemicals, Dubai, UAE). Cells were then postfixed in 2% osmium tetroxide (Agar Scientific, Stansted, United Kingdom) for 1 h at room temperature. Dehydration was performed by incubating the sample in increasing ethanol concentrations and then in pure propylene oxide. During dehydration, cells were stained with 1% uranyl acetate (Serva, Heidelberg, Germany) in 70% ethanol. Finally, cells were embedded in a mixture of propylene oxide (Electron Microscopy Sciences) and Epon resin (Serva), then in pure Epon resin. After polymerization at 60 °C, 70 nm thick sections were collected on TEM copper grids (Ted Pella Inc., Redding, CA, USA). Electron micrographs were obtained with Morada camera on a JEM 1400 transmission electron microscope at 80 kV (JEOL Co., Tokyo, Japan) in the Laboratory of Electron Microscopy Core Facility, Nencki Institute of Experimental Biology, Polish Academy of Sciences, Warsaw, Poland.

### 4.8. Statistical Analysis

Statistical analyses were performed using GraphPad Prism Software (San Diego, CA, USA) (https://www.graphpad.com/scientific-software/prism/). Student’s *t*-Test was used to evaluate the statistical significance of differences between the numbers of Sec7-mRFP puncta in transformed yeast cells. One-way ANOVA was used to evaluate the statistical significance of differences between the fractions of cells with the indicated types of Golgi ultrastructure. Data are presented as means ±standard deviations. The differences are considered significant where *p* < 0.05.

## Figures and Tables

**Figure 1 ijms-22-00914-f001:**
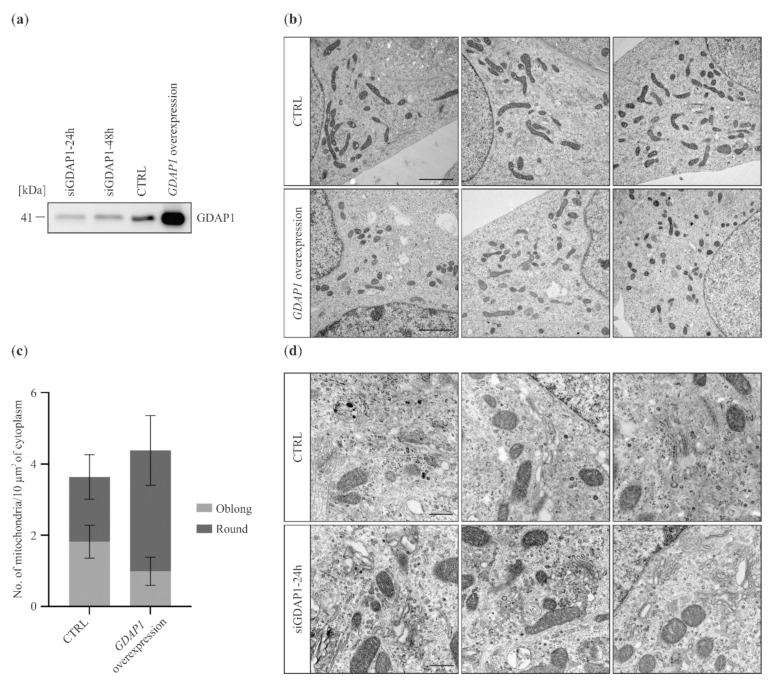
GDAP1 levels in SH-SY5Y cells determines the morphology of mitochondria and the Golgi. (**a**) The level of GDAP1 following 24 or 48 h of siRNA silencing, and after GDAP1 overproduction. Western blot analysis of total cell extracts using an anti-GDAP1 antibody. (**b**) Electron micrographs presented mitochondrial morphology in cells overexpressing *GDAP1*. Scale bar 2 µm. (**c**) Quantification of results from (**b**). (**d**) Morphology of the Golgi following *GDAP1* gene silencing in SH-SY5Y cells on electron micrographs. Scale bar 500 nm.

**Figure 2 ijms-22-00914-f002:**
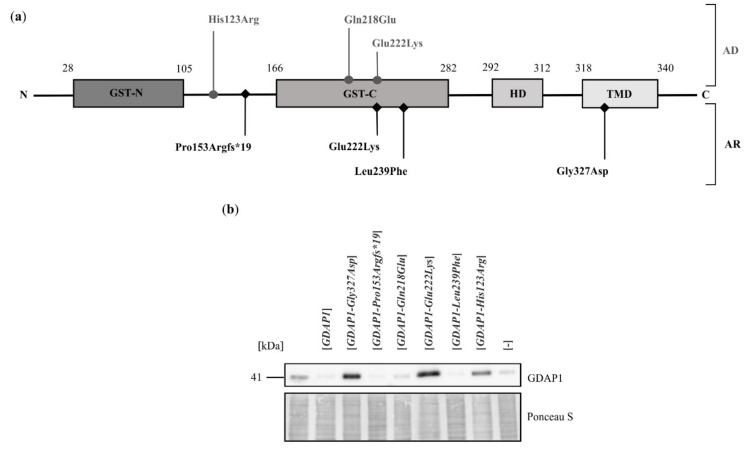
Expression of various *GDAP1* alleles in HeLa cells affects the level of endogenous GDAP1. (**a**) Schematic representation of GDAP1 structure including amino acid residue substitutions that result from the *GDAP1* variants of autosomal dominant (AD; above schematic) or autosomal recessive (AR; below schematic) modes of inheritance. (**b**) Western blot analysis using anti-GDAP1 antibody. Shown are total cell extracts from non-transfected HeLa cells (first lane from left), HeLa cells transfected with empty vector ([-]) or HeLa cells transfected with vectors carrying the indicated *GDAP1* variants. The samples were normalized for protein loading using Ponceau S staining.

**Figure 3 ijms-22-00914-f003:**
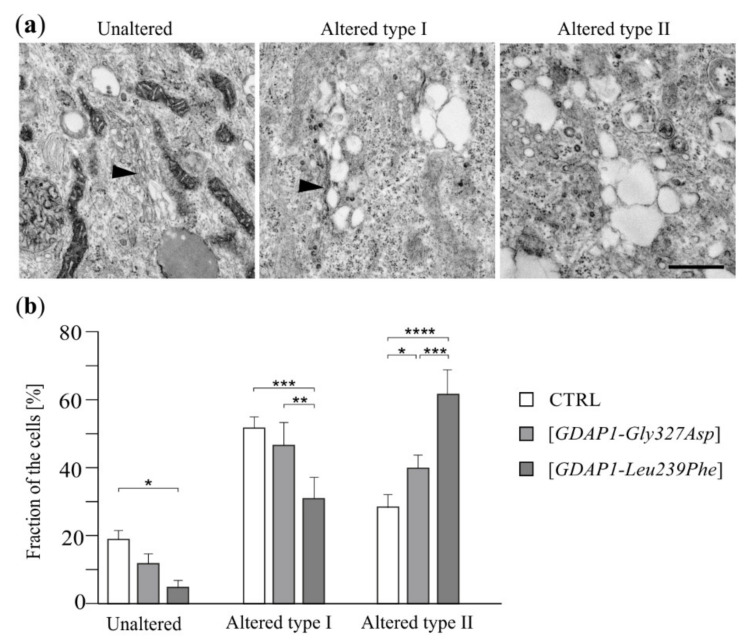
Expression of mutant *GDAP1* variants affects the morphology of the Golgi in HeLa cells. (**a**) Transmission electron microscope analysis of the Golgi in non-transfected HeLa cells (CTRL) or in HeLa cells transfected with a vector carrying *GDAP1-Gly327Asp* or *GDAP1-Leu239Phe* alleles. Electron micrographs of representative examples of Golgi structures observed in HeLa cells. Black arrowheads indicate the cisternae of the Golgi. Scale bar 750 nm. (**b**) Fraction of cells [as percentage of counted cells] with types of Golgi ultrastructure indicated. Statistical analysis was performed on data from three independent experiments using one-way ANOVA and Bonferroni’s correction: * *p* < 0.05, ** *p* < 0.01, *** *p* < 0.001, **** *p* < 0.0001.

**Figure 4 ijms-22-00914-f004:**
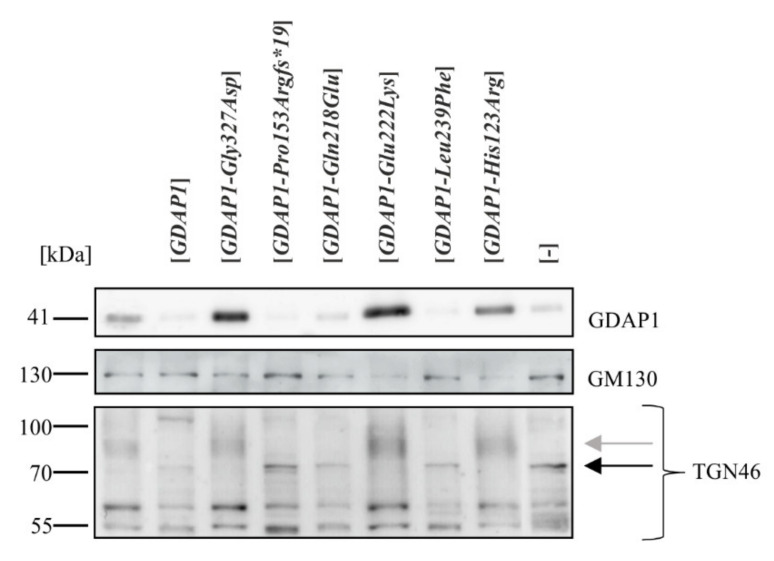
The level of GDAP1 protein correlates with changes in the pattern of TGN46 modifications. Western blot analysis of total cell extracts from not transfected HeLa cells (far left lane), transfected with empty vector ([-]) or vectors bearing variants of *GDAP1* (as indicated). Western blots performed using antibodies against GDAP1 or against the Golgi proteins GM130 and TGN46. The black arrow indicates immature TGN46, the grey arrow shows mature, modified forms.

**Figure 5 ijms-22-00914-f005:**
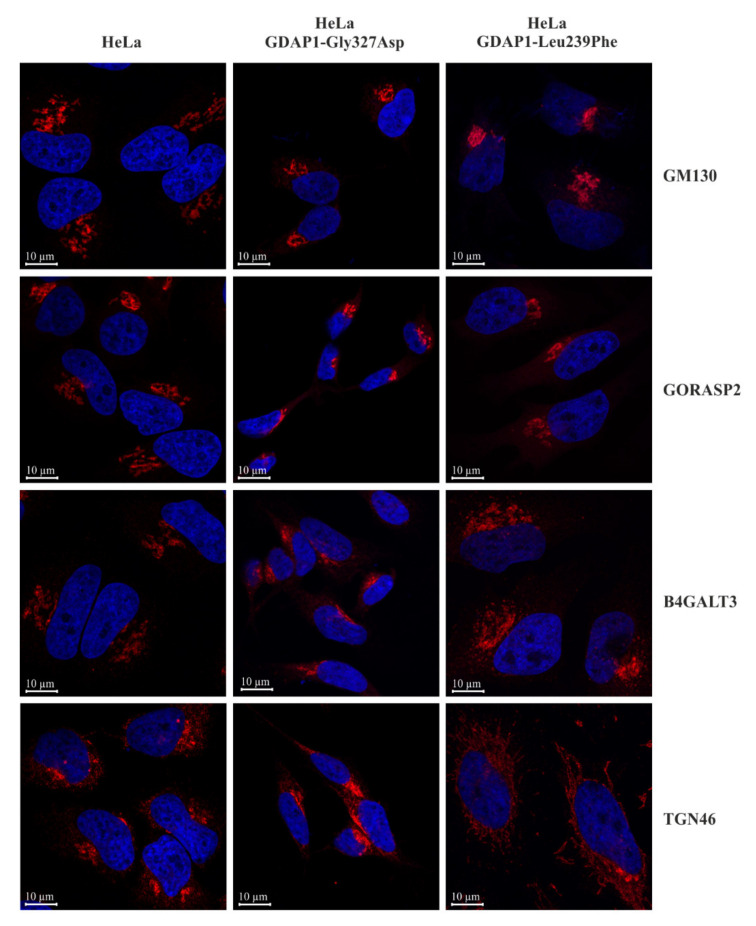
The localization of TGN46, a protein of the *trans*-Golgi network, is altered in HeLa cells expressing the *GDAP1-Leu239Phe* allele. The confocal microscopy images show the localization of GM130, GORASP2, B4GALT3 and TGN46 proteins in the control HeLa cells or in HeLa cells expressing the *GDAP1-Gly327Asp* or *GDAP1-Leu239Phe* alleles. All proteins were visualized by indirect immunofluorescence. Nuclear DNA was stained with DAPI.

**Figure 6 ijms-22-00914-f006:**
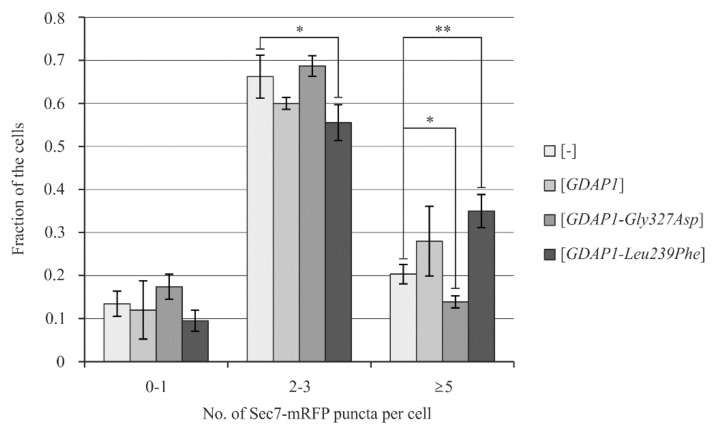
The expression of *GDAP1* variants in yeast cells influences localization of Sec7-mRFP, the *trans*-Golgi network marker protein. Yeast cells carrying the *SEC7-mRFP*-containing plasmid were observed using confocal microscopy. The number of Sec7-mRFP puncta in a single cell was determined. Error bars represent standard deviation for three repeats. Statistical analysis was performed on data from three independent experiments using t-Test. * *p*-value < 0.05 ** *p*-value < 0.01.

**Figure 7 ijms-22-00914-f007:**
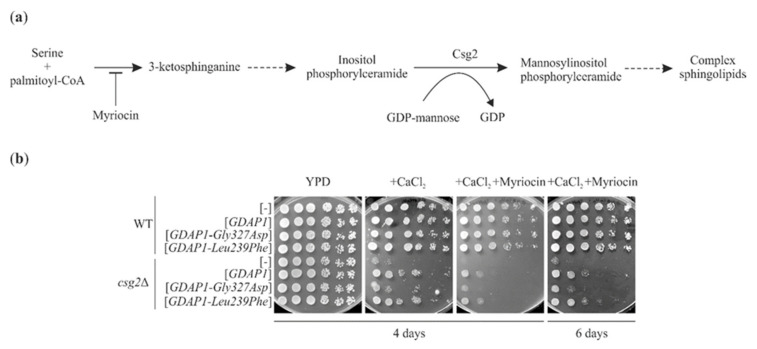
Expression of *GDAP1* or the *GDAP-Leu239Phe* allele suppresses the sensitivity of the *csg2*Δ mutant to Ca^2+^ ions even in the presence of myriocin. (**a**) Schematic representation of the sphingolipid biosynthesis pathway. The step inhibited by myriocin is indicated. (GDP) guanosine diphosphate. (**b**) The growth of wild-type and *csg2*Δ mutant strains transformed with empty vector ([-]), or vectors carrying the cDNA of *GDAP1*, ([*GDAP1*]) or *GDAP1* variants ([*GDAP1-Gly327Asp*] and [*GDAP1-Leu239Phe*]) was compared on YPD media or YPD containing 0.4 M calcium chloride (+CaCl_2_) or 0.4 M CaCl_2_ and 1 µM myriocin (+CaCl_2_ + Myriocin).

**Figure 8 ijms-22-00914-f008:**
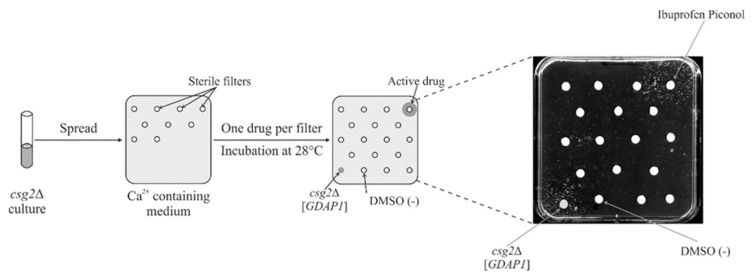
Ibuprofen piconol is a chemical suppressor of the *csg2*Δ growth defect on media supplemented with Ca^2+^ ions. The *csg2*Δ cells were plated on YPD + 0.5 M CaCl_2_. Compounds from the Prestwick library were applied on filter discs (2 µL of 10 mM solution in DMSO). DMSO, a solvent, was used as a negative control. The *csg2*Δ [*GDAP1*] strain was spotted as a positive control. Plates were incubated for 3–5 days at 28 °C.

**Table 1 ijms-22-00914-t001:** Plasmids used in this study.

Name	Description	Source
**Yeast Plasmids:**		
p425-P_TDH3_	2µ; *LEU2*	[53]
p425-P_TDH3_-*GDAP1*	2µ; *LEU2; GDAP1*	[16]
p425-P_TDH3_-*GDAP1m2*	2µ; *LEU2; GDAP1* c.980G > A; p.Gly327Asp	[16]
p425-P_TDH3_-*GDAP1m5*	2µ; *LEU2; GDAP1* c.715C > T; p.Leu239Phe	[16]
Sec7-mRFP	*URA3; SEC7-mRFP*	[31]
Sed5-mRFP	*URA3; SED5-mRFP*	[31]
**Mammalian Expression Plasmids:**		
pCMV6-XL5-*GDAP1*	Human Untagged Clone *GDAP1* cDNA	OriGene
pCMV6-XL5-*GDAP1m1*	*GDAP1* c.456delC; p.Pro153Argfs*19	This study
pCMV6-XL5-*GDAP1m2*	*GDAP1* c.980G > A; p.Gly327Asp	This study
pCMV6-XL5-*GDAP1m3*	*GDAP1* c.652C > G; p.Gln218Glu	This study
pCMV6-XL5-*GDAP1m4*	*GDAP1* c.664G > A; p.Glu222Lys	This study
pCMV6-XL5-*GDAP1m5*	*GDAP1* c.715C > T; p.Leu239Phe	This study
pCMV6-XL5-*GDAP1m6*	*GDAP1* c.368A > G; p.His123Arg	This study
pIRES2-AcGFP1	Bicistronic vector with GFP	TAKARA Bio
pIRES2-AcGFP1-*GDAP1*	GDAP1 WT	This study
pIRES2-AcGFP1-*GDAP1m1*	*GDAP1* c.456delC; p.Pro153Argfs*19	This study
pIRES2-AcGFP1-*GDAP1m2*	*GDAP1* c.980G > A; p.Gly327Asp	This study
pIRES2-AcGFP1-*GDAP1m3*	*GDAP1* c.652C > G; p.Gln218Glu	This study
pIRES2-AcGFP1-*GDAP1m4*	*GDAP1* c.664G > A; p.Glu222Lys	This study
pIRES2-AcGFP1-*GDAP1m5*	*GDAP1* c.715C > T; p.Leu239Phe	This study
pIRES2-AcGFP1-*GDAP1m6*	*GDAP1* c.368A > G; p.His123Arg	This study

## Data Availability

The data presented in this study are available on request from the corresponding author. The data are not publicly available due to privacy.

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
