# Peer review of "Mutations in GDAP1 Influence Structure and Function of the Trans-Golgi Network"

_ijms, 2021, doi:10.3390/ijms22020914_

Round 1

Reviewer 1 Report

The study suggested an important role of the Golgi apparatus in the pathology of CMT with GDAP1 mutations. The authors showed that the GDAP1mutants were associated with changes in the trans-Golgi network. They performed drug screening using the yeast CMT-GDAP1 model, and suggested ibuprofen as an encouraging candidate for the treatment of CMT-GDAP1 disease. It seems that this work was novel and useful for understanding and treatment of CMT with GDAP1 mutations. However, following major and minor questions are needed to be addressed.

[2.1. GDAP1 protein levels influence mitochondria and Golgi morphology in SH-SY5Y cells] The authors stated that the effect of GDAP1 silencing on mitochondria is difficult to estimate, as GDAP1 is still present at low levels and/or in some cells. It is confusing whether the meaning of the difficulty of estimation is that the measurement itself was difficult or that it was difficult to observe a significant difference. If Golgi's morphoogy was observed under the GDAP1 silicing condition in Figure 2d, couldn't the morphology of mitochondria be also observed under this condition? It is recommended to include mitochondria observation at GDAP1 silencing and Golgi observation at overexpression, whether substantial differences are observed or not.

[L172-183, Figure 2] In the case of transfection with vector bearing wild-type GDAP1, shouldn't it be overexpressed as shown in Figure 1a? ln addition, should there be no change in the case of transfection with vector only? Or, does it seem that this case may be used as control compare to other experiments? The western blot showed total GDAP1 protein including endogeneous and exogeneous proteins in Figure 2b. Therefore, it seems to be insufficient to conclude that each corresponding allele regulates up or down expression of the endodogeneous GDAP1.

Minor points

[L13, 31] Charcot-Marie-Tooth (CMT) disease > Charcot-Marie-Tooth disease (CMT)

[L100-113] It seems that the last paragraph of the “Introduction” is too long. In particular, it is recommended to delete following sentence: What is more, the drugs that were selected in our screen 110 may be further developed and tested for efficient CMT-GDAP1 therapy.

[Materials and Methods] Provide City and Country names for the commercial suppliers (such as Sigma-Aldrich, Thermo Fisher Scientific, Lipocalyx, Lab Empire, Qiagen, OriGene, TAKARA, Vazyme biotech, Abcam, and Proteintech), when they were referred first time.

[Line 532] For (Sigma), Sigma or Sigma-Aldrich?

[Line 585] Provide web address or reference for the GraphPad Prism Software.

Author Response

We highly appreciate all efforts of the Reviewer to improve our manuscript, both in general and detailed points. We would like to thank for all advices and suggestions. We have addressed all the points indicated by the Reviewer 1 as follows.

  1. [2.1. GDAP1 protein levels influence mitochondria and Golgi morphology in SH-SY5Y cells] The authors stated that the effect of GDAP1 silencing on mitochondria is difficult to estimate, as GDAP1 is still present at low levels and/or in some cells. It is confusing whether the meaning of the difficulty of estimation is that the measurement itself was difficult or that it was difficult to observe a significant difference. If Golgi's morphoogy was observed under the GDAP1 silencing condition in Figure 2d, couldn't the morphology of mitochondria be also observed under this condition? It is recommended to include mitochondria observation at GDAP1 silencing and Golgi observation at overexpression, whether substantial differences are observed or not.

In SH-SY5Y cells there is no evident differences in mitochondria morphology upon GDAP1 silencing. We explain this result by the fact that the silencing is not total and we cannot estimate the GDAP1 level in the single cell. We cannot discriminate between cells with significant and without GDAP1 silencing to be sure that observed effect on mitochondria morphology is connected to a low GDAP1 protein level.

We addressed this point changing text in lines 140-143.

We did not present the result for Golgi morphology after GDAP1 overexpression, because we were not able to observe Golgi cristae structures in any observed cells. In this case, we are not sure where this result came from; is effect of GDAP1 overexpression or problems with microscopic technique.

  1. [L172-183, Figure 2] In the case of transfection with vector bearing wild-type GDAP1, shouldn't it be overexpressed as shown in Figure 1a? ln addition, should there be no change in the case of transfection with vector only? Or, does it seem that this case may be used as control compare to other experiments? The western blot showed total GDAP1 protein including endogeneous and exogeneous proteins in Figure 2b. Therefore, it seems to be insufficient to conclude that each corresponding allele regulates up or down expression of the endodogeneous GDAP1.

In the Figure 1 there are different cells than in the figure 2 and HeLa cells (Figure 2) are characterized by the very low level of the GDAP1 protein compared to SH-SY5Y cells.

We addressed this point in discussion lines 361-373.

We repeated this experiment several times and the observed GDAP1 protein level after transfection of empty vector fluctuates from experiment to experiment however the tendency of expression of various GDAP1 alleles was the same.

If requested we can provide results of these experiments.

  1. Minor points

[L13, 31] Charcot-Marie-Tooth (CMT) disease > Charcot-Marie-Tooth disease (CMT)

[L100-113] It seems that the last paragraph of the “Introduction” is too long. In particular, it is recommended to delete following sentence: What is more, the drugs that were selected in our screen 110 may be further developed and tested for efficient CMT-GDAP1 therapy.

[Materials and Methods] Provide City and Country names for the commercial suppliers (such as Sigma-Aldrich, Thermo Fisher Scientific, Lipocalyx, Lab Empire, Qiagen, OriGene, TAKARA, Vazyme biotech, Abcam, and Proteintech), when they were referred first time.

[Line 532] For (Sigma), Sigma or Sigma-Aldrich?

[Line 585] Provide web address or reference for the GraphPad Prism Software.

All minor points were addressed in the text and link to GraphPad Prism software was added (line 575).

Reviewer 2 Report

The authors have investigated some GDAP1 mutations in commercial cell lines and in yeast. They have observed that the p.Leu239Phe may alter the trans-Golgi network. Using the yeast model csg2∆ that leads to the suppression of the calcium sensitivity, the authors demonstrated that p.Leu239Phe improves the growth of the csg2∆ mutant. After applying the Prestwick library drug’s, two molecules were identified as suppressors of the csg2∆ growth defect: cyclosporine A and ibuprofen piconol.

Comments:

TITLE & ABSTRACT

I do not think the authors have demonstrated that function of TGN is impaired. Moreover, they have observed the changes on the structure/expression only with one mutation. In sum, the title as well as the abstract should be more adjusted to the obtained findings.

INTRODUCTION

Regarding the role of GDAP1 in the calcium homeostasis, the work focused on the junctophilin as a modifier gene of GDAP1 is not included (Pla-Martín et al. Hum Mol Genet 2015; 24: 213-19).

RESULTS

  • Why do the authors work with the GDAP1 p.Pro153Argfs* mutation if this has not been described in patients? There are several STOP mutations associated with CMT such as p.Tyr124* or p.Arg125*.
  • Regarding nomenclature of mutations, changes on the protein must be named with a previous “p”: p.Pro153Argfs* instead of Pro153Argfs*. Changes on protein do not need italics (only changes on DNA).
  • Figure 4 is not clear. To observe the commented changes in TGN46 is quite difficult. For all the mutations, is this pattern altered in the same way?
  • Figure 5 (immunofluorescence), the GDAP1 p.Leu239Phe mutation seems to alter the expression pattern for TGN46. Why is the pattern of BAGALT3 not altered? This is also a TGN marker. I see pictures quite similar compared with the control. What happens with other mutations? Why does the authors only study p.Gly327Asp and p.Leu239Phe?
  • Figure 6 (yeast). To conclude that GDAP1 mutations affect TGN, other additional mutations should be investigated. All the findings are based on only one GDAP1 change.
  • Figure 7. The csg2∆ mutant phenotype’s is improved expressing the p.Leu239Phe mutation. The csg2∆ mutant is characterized by the suppression of the calcium sensitivity. Which is the novelty of this experiment? It is well established that GDAP1 plays a role in calcium homeostasis.

DISCUSSION

Discussion must be rewritten according of the modifications carried out in the previous sections. Moreover, the authors conclude that TGN is impaired due to GDAP1 mutations and hence, intracellular trafficking may be damaged, which is a new pathomechanism for GDAP1-associated neuropathies. In this case, distinct CMT types are known in which trafficking is impaired such as those caused by mutations in MORC2 (Sancho et al. Hum Mol Genet 2019; 28: 1629-44), SH3TC2 (Gouttenoire et al. Glia 2013; 61: 1041-51) or DNM2 (Bitoun et al. Hum Mut 2009; 30: 1419-27).

Author Response

We thank the Reviewer for his/her time an insightful comments that have made improvements to this manuscript. We have addressed all the points indicated by the Reviewer 2 as follows.

  1. TITLE & ABSTRACT

I do not think the authors have demonstrated that function of TGN is impaired. Moreover, they have observed the changes on the structure/expression only with one mutation. In sum, the title as well as the abstract should be more adjusted to the obtained findings.

We studied not only changes in Golgi morphology upon GDAP1-Leu239Phe mutant allele expression but also in SH-SY5Y cells with GDAP1 silencing and moreover, Western blot analysis were performed for six different GDAP1 mutations and in case of three, we observed alterations in TGN46 modification pattern. Therefore we decide to use more general title, not regarding only one particular mutation.

  1. INTRODUCTION

Regarding the role of GDAP1 in the calcium homeostasis, the work focused on the junctophilin as a modifier gene of GDAP1 is not included (Pla-Martín et al. Hum Mol Genet 2015; 24: 213-19).

 We added the suggested information in line 60.

  1. RESULTS
  • Why do the authors work with the GDAP1 p.Pro153Argfs* mutation if this has not been described in patients? There are several STOP mutations associated with CMT such as p.Tyr124* or p.Arg125*.

This mutation is representative for all mutation resulting in GDAP1 truncation.

  • Regarding nomenclature of mutations, changes on the protein must be named with a previous “p”: p.Pro153Argfs* instead of Pro153Argfs*. Changes on protein do not need italics (only changes on DNA).

In text we used the name of alleles which are based on amino acid substitution observed on protein level. We decided to use these names not to confuse the readers and to simplified the text.

  • Figure 4 is not clear. To observe the commented changes in TGN46 is quite difficult. For all the mutations, is this pattern altered in the same way?

We addressed this point in lines 224 - 226 by changing sentence as follows: Western blot analysis revealed that the expression of some GDAP1 mutant alleles resulted in a drop in the endogenous GDAP1 level, and this correlates with a change to the pattern of posttranslational modification of TGN46. In cells expressing GDAP1, GDAP1-Pro153Argfs*19, GDAP1-Gln218Glu, and GDAP1-Leu239Phe, TGN46 did not fully mature; the slower migrating forms, shown previously to be sialylated forms, were absent.

  • Figure 5 (immunofluorescence), the GDAP1 p.Leu239Phe mutation seems to alter the expression pattern for TGN46. Why is the pattern of BAGALT3 not altered? This is also a TGN marker. I see pictures quite similar compared with the control. What happens with other mutations? Why does the authors only study p.Gly327Asp and p.Leu239Phe?

Western blot analysis did not reveal any specific pattern of B4GALT3 and we did not observe any changes between lines. Thus, the immunofluorescence is in agreement with Western blot analysis.

The B4GALT3 protein and other Golgi apparatus markers shuttle between ER and within Golgi apparatus and they are predominantly present in cis-, medium- or trans-Golgi [Tu and Banfield, Cellular and Molecular Life Sciences, 2010]

Based on presented Western blot analysis in figure 4 in which we divided mutations into two groups, we choose two mutations in the GDAP1 gene (one from each group) to be studied by immunofluorescence. The amino acid substitution at residue 239 (p.Leu239Phe) is the most common one [Kabzińska et al., Neurogenetics, 2010], and the second is only known missense GDAP1 mutation resulting in disturbed GDAP1 anchoring to mitochondrial membrane [Kabzińska et al., Neurogenetics, 2011].

We addressed this point in lines 233 – 237.

  • Figure 6 (yeast). To conclude that GDAP1 mutations affect TGN, other additional mutations should be investigated. All the findings are based on only one GDAP1 change.

We investigated the same mutations as in HeLa cells (fig. 5) to observed if these mutation results in the similar changes as in mammalian cells.

We clarified this point in line 258.

  • Figure 7. The csg2∆ mutant phenotype’s is improved expressing the p.Leu239Phe mutation. The csg2∆ mutant is characterized by the suppression of the calcium sensitivity. Which is the novelty of this experiment? It is well established that GDAP1 plays a role in calcium homeostasis.

Novelty of this experiment is the fact that we tested the effect of myriocin treatment on ability of GDAP1 suppression of csg2∆ - calcium sensitivity, mechanism of which is unknown. The obtained result indirectly indicates that ongoing sphingolipid synthesis is not required for GDAP1 function in yeast cells. This is a step for further understanding of the GDAP1 molecular function. The SOCE pathway is not presented in yeast cells thus the mechanism of calcium homeostasis regulation in yeast cells cannot be the same as in mammals cells.

  1. DISCUSSION

Discussion must be rewritten according of the modifications carried out in the previous sections. Moreover, the authors conclude that TGN is impaired due to GDAP1 mutations and hence, intracellular trafficking may be damaged, which is a new pathomechanism for GDAP1-associated neuropathies. In this case, distinct CMT types are known in which trafficking is impaired such as those caused by mutations in MORC2 (Sancho et al. Hum Mol Genet 2019; 28: 1629-44), SH3TC2 (Gouttenoire et al. Glia 2013; 61: 1041-51) or DNM2 (Bitoun et al. Hum Mut 2009; 30: 1419-27).

We added the suggested information in lines 382 – 384 and 421 - 426.

Round 2

Reviewer 1 Report

It seems that the author addressed properly for all the comments by the reviewer.

Reviewer 2 Report

The authors have addressed all the issues raised by the reviewer. The paper is now suitable to be published in the International Journal of Molecular Sciences